# Expression and Functional Analyses of Dlk1 in Muscle Stem Cells and Mesenchymal Progenitors during Muscle Regeneration

**DOI:** 10.3390/ijms20133269

**Published:** 2019-07-03

**Authors:** Lidan Zhang, Akiyoshi Uezumi, Takayuki Kaji, Kazutake Tsujikawa, Ditte Caroline Andersen, Charlotte Harken Jensen, So-ichiro Fukada

**Affiliations:** 1Project for Muscle Stem Cell Biology, Graduate School of Pharmaceutical Sciences, Osaka University, 1–6 Yamadaoka, Suita, Osaka 565–0871, Japan; 2Laboratory of Molecular and Cellular Physiology, Graduate School of Pharmaceutical Sciences, Osaka University, 1–6 Yamadaoka, Suita, Osaka 565–0871, Japan; 3Muscle Aging and Regenerative Medicine, Tokyo Metropolitan Institute of Gerontology, Itabashi-ku, Tokyo 173–0015, Japan; 4Laboratory of Molecular and Cellular Cardiology, Department of Clinical Biochemistry and Pharmacology, Odense University Hospital, Winsloewparken 21 3rd, 5000 Odense C, Denmark; 5Clinical Institute, University of Southern Denmark, Winsloewparken 21 3rd, 5000 Odense C, Denmark

**Keywords:** Dlk1, muscle regeneration, muscle stem cells, mesenchymal progenitors

## Abstract

Delta like non-canonical Notch ligand 1 (*Dlk1*) is a paternally expressed gene which is also known as preadipocyte factor 1 (*Pref−1*). The accumulation of adipocytes and expression of Dlk1 in regenerating muscle suggests a correlation between fat accumulation and Dlk1 expression in the muscle. Additionally, mice overexpressing Dlk1 show increased muscle weight, while Dlk1-null mice exhibit decreased body weight and muscle mass, indicating that Dlk1 is a critical factor in regulating skeletal muscle mass during development. The muscle regeneration process shares some features with muscle development. However, the role of Dlk1 in regeneration processes remains controversial. Here, we show that mesenchymal progenitors also known as adipocyte progenitors exclusively express Dlk1 during muscle regeneration. Eliminating developmental effects, we used conditional depletion models to examine the specific roles of Dlk1 in muscle stem cells or mesenchymal progenitors. Unexpectedly, deletion of Dlk1 in neither the muscle stem cells nor the mesenchymal progenitors affected the regenerative ability of skeletal muscle. In addition, fat accumulation was not increased by the loss of Dlk1. Collectively, Dlk1 plays essential roles in muscle development, but does not greatly impact regeneration processes and adipogenic differentiation in adult skeletal muscle regeneration.

## 1. Introduction

The delta like non-canonical Notch ligand 1 (*Dlk1*)-iodothyronine deiodinase 3 (*Dio3*) gene cluster is paternally expressed and involved in metabolism switching, stem cell maintenance, and cell differentiation [1,2]. The best known phenotype of the accelerated expression of the *Dlk1-Dio3* gene cluster is skeletal muscle hypertrophy in sheep, known as *callipyge* sheep [3]. Additionally, mice overexpressing Dlk1 show increased muscle weight [4], while Dlk1-null mice exhibit growth retardation including decreased body weight and skeletal muscle mass [5,6,7], indicating that Dlk1 plays critical roles in skeletal muscle biology. In adult skeletal muscle, including in myofibers, Dlk1 expression is rarely detected [8]. However, the reappearance of Dlk1 expression is observed in Becker and Duchenne muscular dystrophies (DMD), in which the muscle regeneration and degeneration cycles are repeated [8]. Dlk1 expression is also detected in adult murine regenerating muscle, suggesting that Dlk1 is an important factor for muscle regeneration or degeneration.

Muscle regeneration ability depends on muscle stem cells marked with Pax7 [9], known as muscle satellite cells (MuSCs) [10,11,12,13]. A group of skeletal muscle-specific basic helix-loop-helix transcription factors including MyoD, Myf5, myogenin, and Mrf4 (also known as Myf6) are critical for MuSC activation, proliferation, and differentiation as well as skeletal muscle development [14,15]. Using conditional *Dlk1*-null mice inactivated by *Myf5*-Cre (*Myf5-Dlk1* cKO), Waddell et al. showed that Dlk1 is necessary for proper skeletal muscle development and regeneration [5]. *Myf5-Dlk1* cKO mice showed a reduced body weight and muscle mass because of their decreased myofiber numbers. Using cardiotoxin-injury model, muscle regeneration experiments revealed impaired regeneration processes in *Myf5-Dlk1* cKO mice, suggesting that muscle regeneration is limited due to the loss of Dlk1 in Myf5-positive myogenic-lineage cells. However, since the recombination activity of *Myf5*-Cre driver mice is active at an early embryonic stage (around E9.0), developmental defects may secondarily affect the regenerative process in the adult skeletal muscle of *Myf5-Dlk1* cKO mice. Andersen et al. also reported a development defect of skeletal muscle in *Dlk1*-null mice [7] but in contrast to *Myf5-Dlk1* cKO mice, they observed accelerated muscle regeneration in *Dlk1*-null mice in knife-cut lesion model, when myogenic program genes were induced. Thus, the role of Dlk1 in the muscle regenerative process remains controversial, and studies eliminating the effect of developmental Dlk1-defects are required.

Dlk1, which is also known as preadipocyte factor 1, has been reported to negatively regulate adipogenesis [16]. Importantly, accompanying the progression of the disease, the accumulation of adipocytes is increased in DMD patients [8], and Dlk1 is re-expressed in skeletal muscle of DMD and other muscular dystrophy patients, suggesting that Dlk1 expression level might be correlated with fat accumulation. Mesenchymal progenitors, also known as fibro/adipogenic progenitors, exhibit adipogenic differentiation ability in both humans and mice [17,18,19]. Although the cells expressing Dlk1 have not been fully characterized, perivascular and interstitial Dlk1-expressing cells were detected in regenerating muscle [8]. Because fat accumulation decreases skeletal muscle integrity, force, and function, numerous studies have been conducted to evaluate the molecular mechanism underlying the adipogenic differentiation of mesenchymal progenitors.

To investigate the requirement for Dlk1 in adult skeletal muscle regeneration, we identified Dlk1-expressing cells during muscle regeneration and depleted *Dlk1* in specific populations using *Pax7^CreERT2/+^* (for MuSCs) or *Pdgfra^CreERT2/+^* (for mesenchymal progenitors) in mice. These mice were used to analyze the roles of Dlk1 only in adult muscle regeneration processes because Cre activity can be controlled by injection of tamoxifen. Interestingly, Dlk1 expression was extremely specific to mesenchymal progenitors in the middle stage of the regeneration processes. However, unexpectedly, neither MuSC-specific conditional *Dlk1*-depletion mice nor mesenchymal progenitor-specific conditional mice showed abnormalities. Increased fat accumulation was not observed in mesenchymal progenitor-specific conditional knockout mice. These results indicate that mesenchymal progenitors exclusively express Dlk1 during muscle regeneration, but that Dlk1 is dispensable for skeletal muscle regeneration, suggesting that the requirement for Dlk1 in skeletal muscle biology is limited to developmental processes.

## 2. Results

### 2.1. Role of Dlk1 in Myogenic Cells during Regeneration

MuSCs are indispensable for muscle regeneration and are marked by the expression of Pax7 [9]. Dlk1 is also expressed in MuSCs during development and remodeling and therefore, to determine the roles of MuSC-Dlk1 in adult skeletal muscle regeneration, we generated *Pax7^CreERT2/+^::Dlk1^flox/flox^* mice (P7-cKO) and compared them to control mice (P7-control: *Pax7^+/+^::Dlk1^flox/flox^*). This system allowed us to examine the role of Dlk1 in adult skeletal muscle regeneration because *Dlk1* depletion is controlled by the timing of tamoxifen (Tm) injection.

Two weeks after Tm injection, muscle regeneration was induced by cardiotoxin (CTX) treatment; the muscles were sampled at 4 or 14 days after CTX injection for the analyses of myotube formation or regenerated myofiber, respectively (Figure 1A). Since the Dlk1 expression is not detected in MuSCs, it is difficult to observe the depletion of *Dlk1* in MuSCs. However, efficient recombination by *Pax7^CreERT2/+^* with Tm was confirmed using Rosa-YFP reporter mice [20]. As shown in Figure 1B, the area of new regenerating myotubes marked by embryonic myosin heavy chain (eMyHC) expression in P7-cKO was comparable to that in P7-control mice. The size of regenerated myofibers observed at 14 days after CTX injection was also similar between P7-cKO and control mice (Figure 1C,D). Thus, in contrast to the results using *Myf5*-Cre mice, specific-depletion of *Dlk1* in adult muscle stem cells showed no apparent phenotypes in myotube formation and myofiber generation.

### 2.2. High Expression of Dlk1 in Mesenchymal Progenitors in the Middle Stage of Regeneration

Next, we characterized *Dlk1* expression during muscle regeneration. Compared to the expression pattern of *MyoD* and *myogenin* in regenerating muscle, *Dlk1* was expressed at a relatively late stage during muscle regeneration (Figure 2A). This result is consistent with those of Andersen et al. [7] who studied regeneration in a knife-cut lesion. We also examined Dlk1 protein expression in tissue sections. Using *Pax7::Rosa-YFP* mice, myogenic-lineage cells were labeled with yellow fluorescent protein (YFP). As observed by reverse transcription polymerase chain reaction (RT-PCR), a large number of Dlk1-expressing cells were detected in regenerating muscle five or seven days after CTX injection, while few cells were positive for Dlk1 at three days after injection (Figure 2B). Additionally, very low levels of Dlk1 staining were detected in myogenic cells (YFP^+^ cells), indicating that the vast majority of Dlk1 expression originates from non-myogenic cells.

Although the characteristics of Dlk1^+^ cells have not been fully identified, similarly to in a previous study [7], Dlk1^+^ cells were found to reside in the interstitium of regenerating muscle (Figure 2B). To determine the type of cell expressing Dlk1, we isolated CD31/45, myogenic, and mesenchymal progenitor (Sca−1^+^CD31^-^CD45^-^) fractions from regenerating muscle five days after CTX injection (Figure 3A) and evaluated the expression of *Dlk1* compared to in positive controls. Each positive control (*F4/80*; for macrophages, *Pax7*; for myogenic cells, and *Pdgfrα*; for mesenchymal progenitors) was specifically expressed in each fraction. Using these samples, we found that the *Dlk1* expression was highly enriched in mesenchymal progenitors (Figure 3B). We obtained similar results for samples isolated from regenerating muscle at seven days after CTX injection.

Next, we examined the protein expression of Dlk1 in regenerating muscles using tissues sections. Consistent with the results of mRNA expression, Dlk1 staining overlapped with Pdgfrα-staining (Figure 3C). Collectively, these data indicate that mesenchymal progenitors almost exclusively express Dlk1 in the middle stage of adult skeletal muscle regeneration.

### 2.3. Effect of Dlk1-Depletion in Mesenchymal Progenitors during Regeneration

Next, we examined the importance of Dlk1 in mesenchymal progenitors during muscle regeneration using *Pdgfra^CreERT2/+^* and *Dlk1^flox/flox^* mice (Figure 4A). First, we confirmed the successful depletion of Dlk1 in Pa-cKO mice (Figure 4B). Efficient recombination (about 80%) by *Pdgfra^CreERT2/+^* with Tm was also confirmed using Rosa-YFP reporter mice. Using these mice, we injected CTX into Pa-cKO or Pa-control mice, which were sacrificed at 14 days after injection. Unexpectedly, the histology of Pa-cKO mice was comparable to that of control mice (Figure 4C). Although Dlk1 inhibits adipogenic differentiation and mesenchymal progenitors have adipogenic differentiation ability, we observed no apparent abnormalities, including the appearance of adipocytes, in Pa-cKO mice compared to in Pa-control mice using Hematoxylin-Eosin staining (Figure 4C) or Oil-red O staining. The diameter of regenerated myofibers in Pa-cKO was also similar to that in control mice, indicating normal muscle regeneration in both Pa-cKO and Pa-control mice (Figure 4D,E). Collectively, these results indicate that Dlk1 expression is remarkably induced in mesenchymal progenitors during muscle regeneration, but also that Dlk1 is dispensable for CTX-induced muscle regeneration.

## 3. Discussion

### 3.1. Dlk1 Expression in Mesenchymal Progenitors

In this study, we observed cell type and time-specific expression of Dlk1 during muscle regeneration processes. Because Dlk1 expression is detected in proliferative hepatoblasts from the fetal liver [21,22] and in cancer cells [23,24], Dlk1 expression may also reflect the proliferative ability of mesenchymal progenitors. However, we observed that suppression of Dlk1 in mesenchymal progenitors did not affect the process of muscle regeneration. Considering the timing of the peak cell number (three days after CTX injection) and the important roles of mesenchymal progenitors during muscle regeneration [25], these results indicate that Dlk1 is dispensable for the function of mesenchymal progenitors.

In agreement with previous reports on knife-cut lesion model [7], Dlk1 expression is rarely observed in myogenic lineage cells of adult regenerating muscle of CTX-injury model. Previously, we reported that doublecortin is expressed specifically in myogenic cells during the middle stage of regenerating muscle [26]. In the middle stage of regeneration, new nascent myofibers become mature myofibers and the environment clearly differs from that in the early stage of regeneration [27]. Presumably, this difference in environment affects the middle stage-specific expression of Dlk1 in mesenchymal progenitors, and Dlk1^+^ mesenchymal progenitors have some roles that differ from Dlk1^-^ mesenchymal progenitor in the early stage of muscle regeneration, although Dlk1 itself has no impact on muscle regeneration process.

### 3.2. Dlk1 and Adipogenic Differentiation

Mesenchymal progenitors are defined by their ability to differentiate into adipocytes [17,18], and ectopic adipocyte accumulation in the skeletal muscle is detected in patients with muscular dystrophy. In contrast, mesenchymal progenitors also promote the process of muscle regeneration [25]. The origin of this duality is currently unclear. Because Dlk1 has anti-adipogenesis effects [28], Dlk1 is a candidate as the regulator of this duality. However, we found that fat accumulation was not accelerated by the loss of Dlk1 in regenerated muscle. Additional studies are necessary to determine the molecular mechanism directing this duality, which may lead to the development of therapeutic agents for muscular dystrophy.

### 3.3. Function of Dlk1 in Myogenic Cells

Inheritance of the *callipyge* sheep phenotype is known as an unusual mode of inheritance known as the *callipyge* phenotype. This phenotype is observed when a mutated allele is passed on only by the father. Thus, double mutated allele sheep exhibit a wild-type phenotype. Gao et al. proposed that a mutation in the maternal allele induces expression of the *miR−379*/*miR−544* cluster, which suppresses Dlk1 expression, as deletion of the maternal *miR−379*/*miR−544* cluster led to muscle hypertrophy following Dlk1 up-regulation [29]. Additionally, transgenic mice expressing ovine *Dlk1* under the muscle specific murine Myosin light chain 3F promoter and 2E enhancer (*Mlc 3F*/*2E*) exhibited muscular hypertrophy [4], indicating that Dlk1 derived from myogenic cells is functionally important for muscle hypertrophy. However, the expression of Dlk1 is rarely detected in normal and regenerating adult skeletal muscle cells including myofibers and satellite cells. In contrast, developmental myogenic cells express high levels of Dlk1 [8,30]. Considering our result and those observed in *Myf5-Dlk1* cKO and *Dlk1*-null mice, the expression pattern and function of Dlk1 is completely different during the developmental and regenerative processes in the skeletal muscle, and Dlk1 is dispensable for myogenic cells in adult intact and regenerating muscle.

## 4. Materials and Methods

### 4.1. Mice

*Pax7^CreERT2/+^* (Stock No: 012476) [31], *Dlk1 ^flox/flox^* (Stock No: 019074) [32], and *Pdgfra^CreERT2/+^* (Stock No: 018280) [33] mice were obtained from Jackson Laboratories (Bar Harbor, ME, USA). Relocation of *Rosa26^EYFP/+^* (Stock No: 006148) [34] from the the Experimental Animal Care and Use Committee at Osaka University (Approval No. 25-9-3, 25 June 2017; 30-1-1, 8 May 2019). To generate *Pax7^CreERT2/+^::Dlk1 ^flox/flox^* or *Pdgfra^CreERT2/+^::Dlk1 ^flox/flox^* mice, male *Pax7^CreERT2/+^::Dlk1 ^flox/+^* or *Pdgfra^CreERT2/+^::Dlk1 ^flox/+^* mice and female *Dlk1 ^flox/+^* were crossed. Mice were injected twice (*Pax7^CreERT2/+^*) or five times (*Pdgfra^CreERT2/+^*) (24 h apart) intraperitoneally with 200–300 µL tamoxifen (20 mg/mL; #T5648, Sigma-Aldrich, St. Louis, MO, USA) dissolved in sunflower seed oil (#S5007, Sigma-Aldrich) containing 5% ethanol. Three animals were housed in one cage designed for six mice and maintained in a controlled environment (temperature of 24 ± 2 °C, humidity of 50 ± 10%) with a 12:12 h light/dark cycle. The mice received sterilized standard chow (DC−8, Nihon Clea, Tokyo, Japan) and water ad libitum. All procedures involving experimental animals were approved by the Experimental Animal Care and Use Committee at Osaka University. Primer pairs for genotyping are as follows: 5′- ACT AGG CTC CAC TCT GTC CTT C and 5′- GCA GAT GTA GGG ACA TTC CAG TG for *Pax7^CreERT2/+^*; 5′- TCA GCC TTA AGC TGG GAC AT and 5′- ATG TTT AGC TGG CCC AAA TG for *Pdgfra^CreERT2/+^*; 5′- AGA TTC CCC CAC CTC CAA C and 5′- TTC CCA AAC TGG ACA TGA GC for *Dlk1 ^flox/flox^*; 5′- AAA GTC GCT CTG AGT TGT TAT, 5′- AAG ACC GCG AAG AGT TTG TC, and 5′-GGA GCG GGA GAA ATG GAT AT G for *Rosa26^EYFP/+^*.

### 4.2. Muscle Injury

Muscles were injured by injecting cardiotoxin from *Naja pallida* (10 µM in saline)(Latoxan, Valence, France) into the hindlimb muscles (tibialis anterior (TA), gastrocnemius (GC), and quadriceps muscle (Qu)) [35].

### 4.3. Preparation and FACS Analyses of Skeletal Muscle-Derived Mononuclear Cells

Mononuclear cells from injured limb muscles (TA, GC, Qu) were prepared using 0.2% collagenase type II (Worthington Biochemical Corp., Lakewood, NJ, USA) as previously described [36]. Lympholyte (Cedarlane Laboratories, Ltd., Burlington, ON, Canada) was used to remove debris according to the manufacturer’s instructions [37].

Mononuclear cells derived from the skeletal muscles were stained with fluorescein isothiocyanate-conjugated anti-CD31, and CD45, phycoerythrin-conjugated anti-Sca−1, and biotinylated-SM/C−2.6 [38] antibodies. Cells were then incubated with streptavidin-labeled allophycocyanin (BD Biosciences, Franklin Lakes, NJ, USA) on ice for 30 min and resuspended in phosphate-buffered saline containing 2% fetal calf serum and 2 µg/mL propidium iodide. Cell sorting was performed using an FACS Aria II^TM^ flow cytometer (BD Immunocytometry Systems, Mountain View, CA, USA). Debris and dead cells were excluded by forward scatter, side scatter, and propidium iodide gating. Data were collected using FACSDiva^TM^ software (BD Biosciences).

### 4.4. Real-Time PCR

Total RNA was extracted from sorted cells using Trizol LS reagent (Thermo Fisher Scientific, Waltham, MA, USA) and a QIAGEN RNeasy Micro Kit according to the manufacturer’s instructions (Hilden, Germany). Total RNA was extracted from uninjured and injured TA muscles using Qiagen Tissue Ruptor Disposable Probes (Nonsterile) (#990890), Qiagen Tissue Ruptor (#9001271), and a Qiagen RNeasy Fibrous Tissue Mini Kit (#74704) according to the manufacturer’s instructions. The isolated total RNA was reverse-transcribed into cDNA using a QuantiTect Reverse Transcription Kit (QIAGEN). Real-time PCR was performed using SYBR Green Universal Mix (#13608700, Roche Diagnostics, Mannheim, Germany) and StepOnePlus Real-Time PCR System (Applied Biosystems, Foster City, CA, USA). Specific forward and reverse primers for optimal amplification in real-time PCR of reverse transcribed cDNAs were as follows: 5′-AAG CAT CCG AGA CAC ACA CA and 5′-GGC AAG ACA TAC CAG GGA GA for mouse F4/80; 5′-GAC GAG TGT CCT TCG CCA AAG TG and 5′-CAA AAT CCG ACC AAG CAC GAG G for mouse Pdgfra; 5′-GTC TGG TTC AGT AAC CGG CGT G and 5′-GGT TAG CTC CTG CCT GCT TA for mouse Pax7; 5′-AGA TTC CCC CAC CTC CAA C and 5′-TTC CCA AAC TGG ACA TGA GC for mouse Dlk1; 5′-CAA CTG CTC TGA TGG CAT GAT G and 5′-AGA TGC GCT CCA CTA TGC TG for mouse MyoD; 5′-TCC CAA CCC AGG AGA TCA TTT G and 5′-ACA ATC TCA GTT GGG CAT GG for mouse myogenin; 5′-TGT CAA GCT CAT TTC CTG G and 5′-TTG GGG GCC GAG TTG GGA TA for mouse Gapdh. Quantitative gene expression data are often normalized to the expression levels of Gapdh.

### 4.5. Immunohistochemistry

For immunohistological analyses, transverse cryosections (6–8 μm) were fixed with 4% paraformaldehyde for 10 min. Anti-embryonic myosin heavy chain (eMyHC) or anti-laminin α2 antibodies were purchased from the Developmental Studies Hybridoma Bank (clone F1.652, Iowa City, IA, USA) or Enzo Life Sciences (clone 4H8–2, Plymouth Meeting, PA, USA), respectively. Goat anti-YFP or goat anti-Pdgfrα antibodies were purchased from R&D Systems (Minneapolis, MN, USA) or SICGEN-Research and Development in Biotechnology Ltd. (#AB0020–200, Carcavelos, Portugal), respectively. Rabbit anti-Dlk1 antibody used in this study was reported previously [7]. For staining with mouse anti-eMyHC antibody, transverse cryosections were fixed in cooled acetone for 10 min, and a MOM kit (Vector Laboratories, Burlingame, CA, USA) was used to block endogenous mouse IgG before the reaction with primary antibodies. The signals were recorded photographically using a BZ-X700 fluorescence microscope, and eMyHC-positive areas and myofiber-diameters were quantified with Hybrid Cell Count software (Keyence, Osaka, Japan). For the quantification, all the area of one section was used.

### 4.6. Statistics

Values were expressed as the means ± SD. Statistical significance was assessed by Student’s *t* test. In comparisons of more than two groups, non-repeated measures analysis of variance followed by the Bonferroni test (vs. control) or Student-Newman-Keuls test (multiple comparisons) were used. A probability of less than 5% (*p* < 0.05) or 1% (*p* < 0.01) was considered statistically significant.

## 5. Conclusions

During muscle regeneration, mesenchymal progenitors almost exclusively express Dlk1 during the middle stage of regeneration processes, but Dlk1 is nonetheless dispensable for muscle regeneration processes. These results suggest the limited importance of Dlk1 in post-developmental processes.

## Figures and Tables

**Figure 1 ijms-20-03269-f001:**
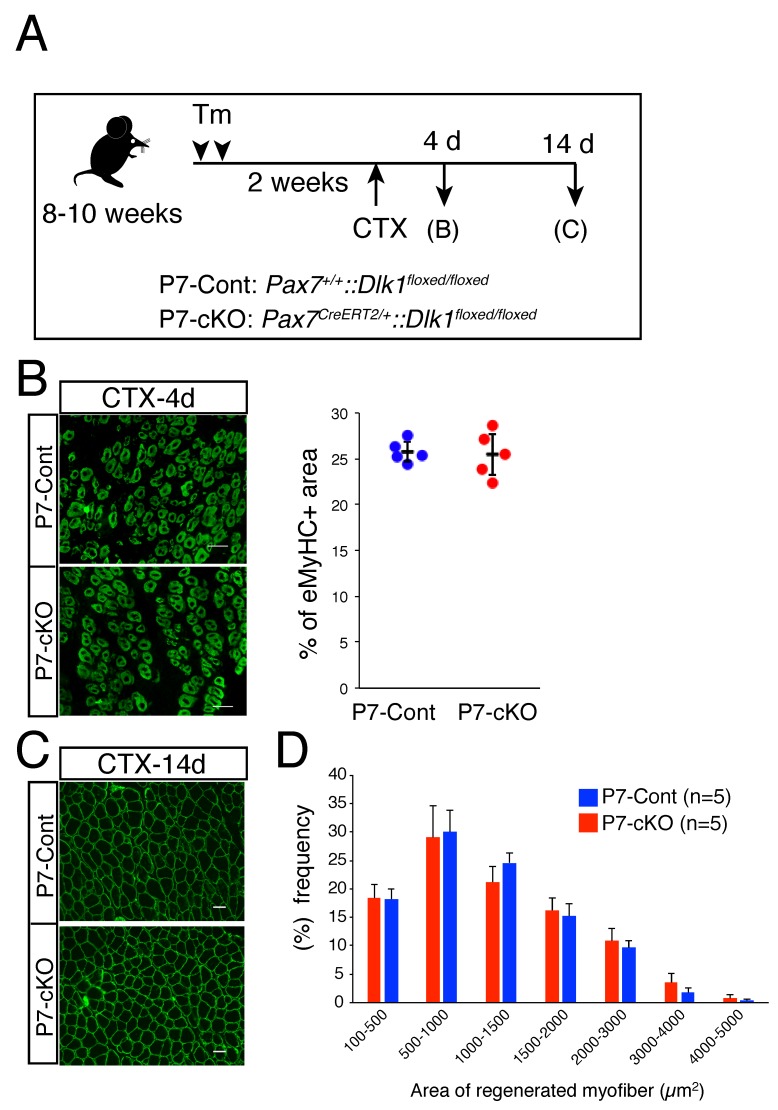
Role of Dlk1 in myogenic lineage cells. (**A**) Cardiotoxin (CTX) and tamoxifen (Tm) time scheme for analysis of the regenerative potential of P7-control and P7-cKO mice. (**B**) Immunostaining of embryonic myosin heavy chain (eMyHC, green) in injured TA muscle 4 days after CTX injection. The y-axis shows the eMyHC^+^ area (percentage) in P7-control (blue circle, *n* = 5) and P7-cKO (red circle, *n* = 5) mice. Error bars indicate the mean with SD. Scale bar: 50 µm. (**C**) Immunostaining of laminin α2 (green) in P7-control and P7-cKO regenerated TA muscle 14 days after CTX injection. Scale bar: 50 µm. (**D**) Graphs indicate the quantitative analyses of the myofiber areas in P7-control (blue bar) and P7-cKO (red bar) mice. The y-axis represents the percentage of each myofiber size range. The x-axis indicates the size of the myofibers. Data show the average of five mice per group from two independent experiments with SD.

**Figure 2 ijms-20-03269-f002:**
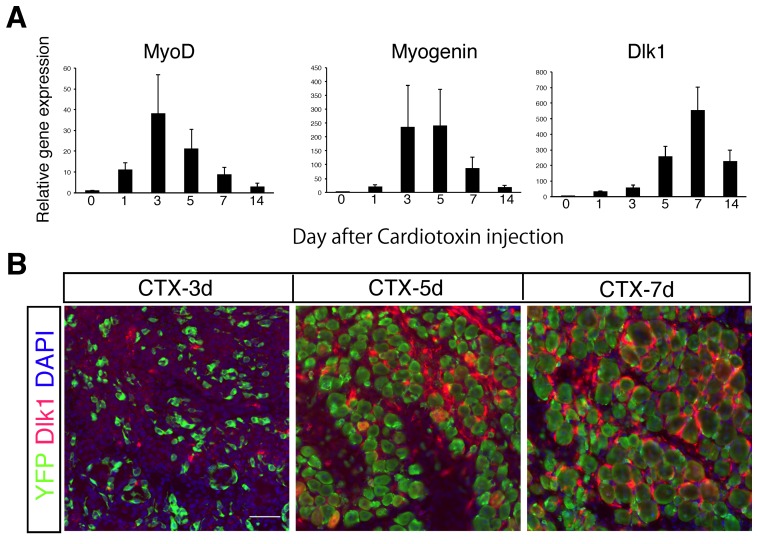
Dlk1 expression during muscle regeneration. (**A**) Relative expression level of MyoD, myogenin, or Dlk1 in intact and regenerating TA muscles. Data show the average of five independent experiments with the SD. (**B**) Immunostaining of Dlk1 (red), myogenic cells (green), and DAPI in regenerating muscle of *Pax7^CreERT2/+^::Rosa-YFP* mice treated tamoxifen on the indicated days after CTX injection. Scale bar: 50 µm.

**Figure 3 ijms-20-03269-f003:**
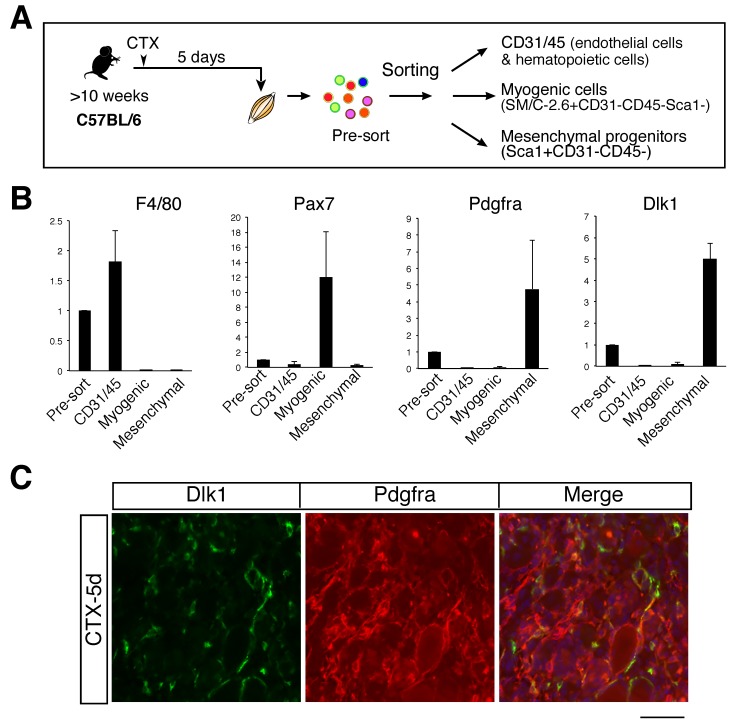
Specific expression of Dlk1 in mesenchymal progenitors. (**A**) Experimental scheme for analyses of Dlk1 in mononuclear cells from regenerating hindlimb muscle. (**B**) Relative mRNA expression levels of Dlk1, macrophage (F4/80), myogenic cells (Pax7), and mesenchymal progenitor marker (Pdgfra) in each cell fraction. Data show the average of three independent sorting experiment with SD. Three mice were used in this study. (**C**) Immunostaining of Dlk1 (green), Pdgfrα (red), and DAPI (blue) in the regenerating TA muscle of C57BL/6 five days after CTX injection. Scale bar: 50 µm.

**Figure 4 ijms-20-03269-f004:**
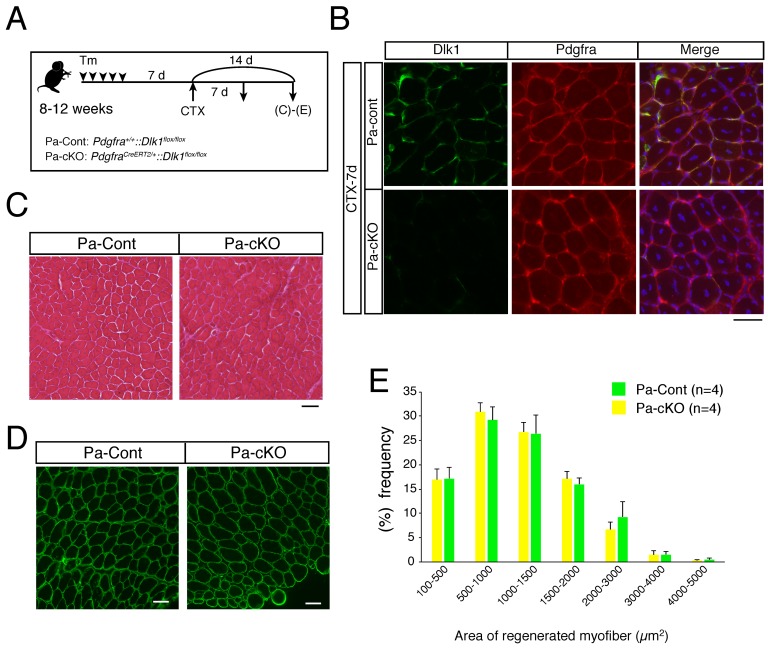
Depletion of Dlk1 in mesenchymal progenitors. (**A**) Cardiotoxin (CTX) and tamoxifen (Tm) time scheme for analysis of the regenerative potential in Pa-control and Pa-cKO mice. (**B**) Immunostaining of Dlk1 (green), Pdgfrα (red), and DAPI (blue) in the regenerating TA muscle of Pa-control and Pa-cKO seven days after CTX injection. Scale bar: 50 µm. (**C**) TA muscle sections of Pa-control and Pa-cKO mice were examined by H&E staining two weeks after CTX injection. Scale bar: 50 µm. (**D**) Immunostaining of laminin α2 (green) in Pa-control and Pa-cKO regenerated TA muscle 14 days after CTX injection. Scale bar: 50 µm. (**E**) Graphs indicate the quantitative analyses of the myofiber areas in Pa-control (green bar) and Pa-cKO (yellow bar) mice. The y-axis shows the percentage of each size of myofibers. The x-axis shows the size of myofibers. Data show the average of four mice per group from two independent experiments with SD.

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
