# Peer review of "Expression and Functional Analyses of Dlk1 in Muscle Stem Cells and Mesenchymal Progenitors during Muscle Regeneration"

_ijms, 2019, doi:10.3390/ijms20133269_

Round 1
Reviewer 1 Report
The paper presented by Lidan Zhang and colleagues is carefully written, data are well presented and results are interesting.
A few issues should be addressed:
- in the Abstract and in the Discussion sections (3.2), it is stated that fat accumulation is not accelerated in Dlk1 ko mice, nor adipogenic differentiation is impaired. However, there is not any direct experimental evidence supporting these statements. For example, by performing a Perilipin 1 (Plin1) staining to visualize adipocytes, or an Oil-Red-O or BODIPY.
- Figure 1B, being n=5, it would be better to plot single values than bars.
- Figure 1 legend, related to 1D, change "Pa-cKO" to "P7-cKO"
- Materials and Methods, 4.2 Muscle injury: you injected TA, gastrocnemius and quadriceps muscles with cardiotoxin to induce muscle regeneration, but then only TA muscles were used for all the experiments, right?
Author Response
- in the Abstract and in the Discussion sections (3.2), it is stated that fat accumulation is not accelerated in Dlk1 ko mice, nor adipogenic differentiation is impaired. However, there is not any direct experimental evidence supporting these statements. For example, by performing a Perilipin 1 (Plin1) staining to visualize adipocytes, or an Oil-Red-O or BODIPY.
-We would like to thank for reviewer’s critical comment. We performed Oil-Red-O staining, but there was no difference between control and Pa-cKO mice. The following description was added in the revised manuscript.
Line 190-193; Although Dlk1 inhibits adipogenic differentiation and mesenchymal progenitors have adipogenic differentiation ability, we observed no apparent abnormalities including the appearance of adipocytes in Pa-cKO mice compared to in Pa-Cont mice using H&E staining (Figure 4C) or Oil-red O staining (data not shown).
- Figure 1B, being n=5, it would be better to plot single values than bars.
In revised version, the original bar graph was replaced with the single plot graph.
- Figure 1 legend, related to 1D, change "Pa-cKO" to "P7-cKO"
We sincerely appreciate reviewer’s careful reading. "Pa-cKO" was changed to "P7-cKO"
- Materials and Methods, 4.2 Muscle injury: you injected TA, gastrocnemius and quadriceps muscles with cardiotoxin to induce muscle regeneration, but then only TA muscles were used for all the experiments, right?
We sincerely apologize for confusing reviewer. Injured TA, GC. Qu muscles were used in Figure 3B. The information was added in the revised Materials and Methods.
Reviewer 2 Report
This is a nice short report demonstrating that Dlk1 is not required in muscle stem cells or mesenchymal progenitors during skeletal muscle regeneration. Overall the work is well done and presented in a logical manner. Further, using mouse models to delete Dlk in loss of function studies provide strong evidence that Dlk1 is dispensable in these cells during regeneration.
I only have 2 minor suggestions:
1. It would be useful to confirm the conditional deletion of Dlk in muscle stem cells as was done for mesenchymal cells in Fig. 4B. With that said, the authors demonstrate that Dlk expression in MuSCs is relatively low, so confirming by staining may be prohibitory – if so, please just comment as such in text. The authors should also mention in text their recombination frequency for both muscle stem cell and mesenchymal targeting schemes.
2. The staining for PDGFRa in sections is suboptimal, especially in 4B, making it difficult to determine cells that are actually positive for this protein. The RNA expression data in 3B strongly supports Dlk expression in sorted mesenchymal progenitors, so the authors are correct in saying Dlk is mainly expressed in MPs. But improving the protein staining, again especially in 4B, is necessary - the Pdgfra stain in Pa-cKO is almost absent, while the Pdgfra stain in Pa-cont non-specifically fills the interstitial space.
Author Response
Reviewer 2
This is a nice short report demonstrating that Dlk1 is not required in muscle stem cells or mesenchymal progenitors during skeletal muscle regeneration. Overall the work is well done and presented in a logical manner. Further, using mouse models to delete Dlk in loss of function studies provide strong evidence that Dlk1 is dispensable in these cells during regeneration.
We sincerely appreciate the Reviewer’s critical reading of out manuscript, constructive comments, and recognizing the importance of our work.
I only have 2 minor suggestions:
1. It would be useful to confirm the conditional deletion of Dlk in muscle stem cells as was done for mesenchymal cells in Fig. 4B. With that said, the authors demonstrate that Dlk expression in MuSCs is relatively low, so confirming by staining may be prohibitory – if so, please just comment as such in text. The authors should also mention in text their recombination frequency for both muscle stem cell and mesenchymal targeting schemes.
We sincerely appreciate reviewer’s comments. We added the following description in the revised manuscript.
Line 101-
Since the Dlk1 expression is not detected in MuSCs, it is difficult to observe the depletion of Dlk1 in MuSCs. But, efficient recombination by Pax7CreERT2/+with Tm was confirmed using Rosa-YFP reporter mice (data not shown) [20].
Line 186-
Efficient recombination (about 80%) by PdgfraCreERT2/+with Tm was also confirmed using Rosa-YFP reporter mice (data not shown).
2. The staining for PDGFRa in sections is suboptimal, especially in 4B, making it difficult to determine cells that are actually positive for this protein. The RNA expression data in 3B strongly supports Dlk expression in sorted mesenchymal progenitors, so the authors are correct in saying Dlk is mainly expressed in MPs. But improving the protein staining, again especially in 4B, is necessary - the Pdgfra stain in Pa-cKO is almost absent, while the Pdgfra stain in Pa-cont non-specifically fills the interstitial space.
We apologize for confusing reviewer. We agree that the Pdgfra-staining was insufficient quality in Pa-cKO. The staining of Pa-cont is correct (it is not non-specific staining). In revised version, the original pictures were replaced with new ones.